# Shedding Light on the Hidden Benefit of *Porphyridium cruentum* Culture

**DOI:** 10.3390/antiox12020337

**Published:** 2023-01-31

**Authors:** Davide Liberti, Paola Imbimbo, Enrica Giustino, Luigi D’Elia, Mélanie Silva, Luísa Barreira, Daria Maria Monti

**Affiliations:** 1Department of Chemical Sciences, University of Naples Federico II, Via Cinthia 4, 80126 Naples, Italy; 2Centre of Marine Sciences, University of Algarve, 8005-139 Faro, Portugal

**Keywords:** microalgae, exopolysaccharides, phycoerythrin, antioxidant activity, anti-inflammatory activity, biocompatibility, wound healing

## Abstract

Microalgae can represent a reliable source of natural compounds with different activities. Here, we evaluated the antioxidant and anti-inflammatory activity of sulfated exopolysaccharides (s-EPSs) and phycoerythrin (PE), two molecules naturally produced by the red marine microalga *Porphyridium cruentum* (CCALA415). *In vitro* and cell-based assays were performed to assess the biological activities of these compounds. The s-EPSs, owing to the presence of sulfate groups, showed biocompatibility on immortalized eukaryotic cell lines and a high antioxidant activity on cell-based systems. PE showed powerful antioxidant activity both *in vitro* and on cell-based systems, but purification is mandatory for its safe use. Finally, both molecules showed anti-inflammatory activity comparable to that of ibuprofen and helped tissue regeneration. Thus, the isolated molecules from microalgae represent an excellent source of antioxidants to be used in different fields.

## 1. Introduction

Microalgae are ubiquitous eukaryotic photosynthetic microorganisms that are able to live in different environments, in single colonies, chains, or groups; depending on the species, their size can vary from a few to hundreds of micrometers [1,2,3]. The biodiversity of microalgae is mainly due to their unique ability to adapt and grow even under unfavorable growth conditions (e.g., extreme temperatures, variable salinity, and low or high light intensity) and to produce a wide range of interesting chemical compounds with novel structures and biological activities [4,5]. Among the microalgae, the red marine microalga *Porphyridium cruentum* could be pointed to as a commercial source of various high-value bioproducts [1], to be recovered from the same culture, in order to make the whole process economically feasible [6,7,8,9,10]. In particular, *P. cruentum* produces sulfated exopolysaccharides (s-EPSs) that are accumulated in a layer surrounding the cytoplasmic membrane. These exopolysaccharides act as a mucilage, because *P. cruentum* is without a well-defined cell wall [11]. They are composed of glucuronic acid and several major neutral monosaccharides, such as D- and L-Gal, D-Glc, D-Xyl, D-GlcA, and sulfate groups. S-EPSs from *P. cruentum* have antioxidant [12], immunomodulatory, anti-inflammatory, hypocholesterolemic, antimicrobial, and antiviral activity [13,14]. S-EPSs from *P. cruentum* also exhibit specific rheological properties that can be exploited in food applications [12,15]. In addition to exopolysaccharides, *P. cruentum* produces a broad range of colored pigments, including chlorophylls, carotenoids, and phycobilins, which are commercially utilized in the food, pharmaceutical, and cosmetic industries [16]. Amongst them, phycoerythrin (PE) is a light-harvesting protein with a structure of (αβ)_6_γ complex and a MW ranging from 240 to 260 kDa. Due to its unique biological properties, PE has gained much attention from the food and pharmaceutical industries and in the molecular biology field [17,18,19,20,21,22]. Here, starting from our recent results [10], a comprehensive study on the biological activities of s-EPSs and purified phycoerythrin was carried out in order to verify if the extraction techniques could affect their biological activities.

## 2. Materials and Methods

### 2.1. Reagents

All solvents, reagents, and chemicals were purchased from Sigma-Aldrich (St Louis, MO, USA).

### 2.2. Biocompounds Isolation

S-EPSs and PE were isolated and purified from the culture of *Porphyridium cruentum* (CCALA415) as previously described [10]. Briefly, at the end of cell growth, the culture was centrifuged to recover s-EPSs in the supernatant. The s-EPSs were precipitated by adding pure ethanol (1:2 *v*/*v*) and centrifuging the sample (12,000× *g*, 30 min, and 4 °C). The supernatant was discarded, and the precipitate was freeze-dried. S-EPSs yield was 300 ± 67 g/L, which corresponds to 0.53 g/g_d.w. biomass_. In the case of PE, a crude aqueous extract was obtained *via* sonication (40% amplitude; 20 min, 30 s on and 30 s off) from the harvested biomass. PE was then isolated *via* a one-step purification procedure as reported by Liberti, up to a purity grade of 4 [10].

### 2.3. Eukaryotic Cell Culture and Biocompatibility Assay

Immortalized human keratinocytes (HaCaT, Innoprot, Derio, Spain) and immortalized murine fibroblasts Balb/c-3T3 (ATCC, Virginia, USA) were cultured in 10% foetal bovine serum in Dulbecco’s modified Eagle’s medium, in the presence of 1% penicillin/streptomycin and 2 mM L-glutamine, in a 5% CO_2_ humidified atmosphere at 37 °C. To verify the biocompatibility of the crude extract of s-EPSs and of purified PE, cells were seeded in 96-well plates at a density of 2 × 10^3^/well and, 24 h after seeding, were incubated with increasing concentrations of the extract/compounds (5 to 75 µg/mL for EPS, 5 to 500 µg/mL of total proteins for crude extracts, and 5 nM to 100 nM for purified PE) for 72 h. At the end of the incubation period, cell viability was assessed with the MTT assay. Cell survival is expressed as the percentage of viable cells in the presence of compounds compared with control cells (represented by the average obtained between untreated cells and cells supplemented with the highest concentration of buffer).

### 2.4. In Vitro Antioxidant Assays

The antioxidant activity of the extract/compounds was tested by measuring their ability to scavenge the free radicals 1,1-diphenyl-2-picrylhydrazyl radical and 2,2′-azinobis-[3-ethylbenzthiazoline-6-sulfonic acid] (DPPH and ABTS, respectively) and to reduce or chelate redox active iron and copper (ferric-reducing antioxidant power (FRAP); iron-chelating activity (ICA), and copper-chelating activity (CCA), respectively). DPPH and FRAP assays were performed following the procedure reported by Rodrigues et al. [23], and ascorbic acid and butylhydroxytoluene (BHT), respectively, were used as positive controls at the same concentrations of the sample under test. The ability of the extract/compounds to scavenge the ABTS radical was assessed as previously reported [24]. The results were compared to a calibration curve obtained using Trolox (6-hydroxy-2,5,7,8-tetramethylchromane-2-carboxylic acid) as the standard. ICA and CCA were determined by measuring the formation of the Fe^2+^-ferrozine complex and by using pyrocatechol violet, respectively, according to the method reported by Megias [25]. EDTA was used as a standard at a final concentration of 100 µg/mL. S-EPS or purified PE was tested between 0.05 and 120 µg/mL and 0.2 and 270 nM, respectively. The results are expressed as IC_50_, i.e., the concentration required to scavenge 50% of the free radical or as the highest percentage achieved.

### 2.5. Determination of Intracellular ROS Levels on Eukaryotic Cell Lines by DCFDA Assay

The protective effect of s-EPSs (from 5 to 75 µg/mL) or purified PE (10 nM) against oxidative stress was measured by determining the intracellular reactive oxygen species (ROS) levels, following the protocol used by Imbimbo [26].

### 2.6. Determination of Intracellular Glutathione Levels (DTNB Assay) and Lipid Peroxidation Levels (TBARS Assay) on Eukaryotic Cell Lines

Intracellular GSH levels and lipid peroxidation levels were measured by following the procedure described by Petruk [27] using 12 µg/mL of s-EPSs or 10 nM of purified PE.

### 2.7. Anti-Inflammatory Activity

The anti-inflammatory activity of the compounds was tested by their ability to inhibit cyclooxygenase-2 (COX-2). S-EPS or purified PE was tested at different concentrations (4 and 167 µg/mL for s-EPSs or 10 and 27 nM for purified PE) using a commercial inhibitory screening assay kit, Cayman test kit-560131 (Cayman Chemical Company, Ann Arbor, MI, USA). Ibuprofen was used as a positive control. Results are expressed as a percentage of inhibition of COX-2.

### 2.8. Wound Healing Assay

Wound healing was assessed with a scratch assay. HaCaT cells were seeded at a cell density of 3 × 10^5^ cells/cm^2^ for 24 h, to allow cells to reach about 95% of confluence. Then, cells were washed with PBS, scratched manually with a 200 μL pipet tip, and incubated with 12 µg/mL of s-EPSs or 10 nM of purified PE. The scratch size was monitored at 0 h and 24 h by acquiring images using optical microscopy (Zeiss LSM 710, Zeiss, Germany) at 10× magnification. The width of the wound was measured by using Zen Lite 2.3 software (Zeiss, Germany). Results are expressed as a reduction of the area (fold) compared with untreated cells.

### 2.9. Statistical Analyses

All the experiments were performed in triplicate. Results are presented as the mean of results obtained after three independent experiments (mean ±SD) and compared by one-way ANOVA according to Bonferroni’s method (post hoc) using GraphPad Prism for Windows, version 6.01 (Dotmatics, California, USA).

## 3. Results

### 3.1. s-Exopolysaccharides Characterization

#### 3.1.1. s-EPSs Biocompatibility on Cell-Based Model

s-EPSs were tested for their biocompatibility on two eukaryotic immortalized cell lines: HaCaT (human keratinocytes) and Balb/c-3T3 (murine fibroblasts). Twenty-four hours after seeding, cells were incubated with increasing amounts of s-EPSs (from 5 to 75 µg/mL). After 72 h of incubation, cell viability was assessed by the MTT assay; cell survival is expressed as the percentage of viable cells in the presence of s-EPSs compared with that of control samples (i.e., untreated cells). The results in Figure 1 show that, under all the experimental conditions, the s-EPSs were fully biocompatible on both the cell lines analyzed.

#### 3.1.2. s-EPSs *In Vitro* Antioxidant Activity

The antioxidant activity of s-EPS was evaluated with different *in vitro* analyses: ABTS, DPPH, FRAP, and iron and copper chelating assays. As shown in Table 1, s-EPSs were not able to scavenge the ABTS and DPPH radicals, whereas a slight but significant activity was observed for the chelation of iron and for ferric ion reduction assays. Both tests are based on the ability to act on iron: the former measures the ability of the compounds under test to bind Fe^2+^, whereas the latter analyzes the ability to reduce Fe^3+^ to Fe^2+^. As for the copper chelating assay, the highest activity reached, at the highest concentration tested, was 9 ± 3%, a value much lower than the one obtained by testing the positive control molecule at the same concentration.

#### 3.1.3. s-EPSs Antioxidant Activity on a Cell-Based Model

The antioxidant activity of s-EPSs was also evaluated on HaCaT cells. For this purpose, cells were incubated with increasing concentrations of s-EPSs (from 5 to 50 µg/mL) for 2 h, and then oxidative stress was induced by UVA irradiation (100 J/cm^2^). Immediately after irradiation, the intracellular ROS levels were measured by using H_2_DCFDA as a probe. For each set of experiments, untreated cells were used as a control. As shown in Figure 2, UVA treatment significantly increased the DCF fluorescence (black bars, *p* <0.001). In the absence of stress, s-EPSs induced a slight but significant increase in the intracellular ROS level (Figure 2, white, dashed grey, and dark grey bars on the left part of the graph). Interestingly, when cells were preincubated with s-EPSs prior to being stressed, only 5 and 12 µg/mL were able to protect the cells from ROS formation (Figure 2, light grey and white bars on the right part of the graph), whereas the higher concentrations had no protective effect. This result is in agreement with those of Giordano et al. [28], as antioxidants act at low concentrations, whereas, at high concentrations, they may work as pro-oxidants. Based on these results, s-EPSs were used at 12 µg/mL for further experiments.

To deeply analyze the protective effect of s-EPSs, the intracellular glutathione levels and lipid peroxidation levels were determined with DTNB and TBARS assays, respectively. In the absence of any treatment, a significant decrease (*p* < 0.01) in GSH levels was observed after UVA exposure (Figure 3A), and s-EPSs (grey bars) were able to inhibit GSH oxidation, thus confirming a protective effect against oxidative stress. As for the TBARS assay, a significant increase (*p* < 0.05) in lipid peroxidation levels was observed after UVA treatment (black bars, Figure 3B), but, notably, this effect was inhibited upon pretreatment with s-EPSs (grey bars). Treatment of the cells with exopolysaccharides did not significantly alter either glutathione or lipid peroxidation levels in the absence of UVA treatment (−). Taken together, the results clearly indicate that s-EPSs are able to protect cells from oxidative damage.

#### 3.1.4. *In Vitro* Anti-Inflammatory Activity of s-EPSs

As inflammation is a condition strictly linked to oxidative stress, the anti-inflammatory activity of s-EPSs was measured by evaluating their capacity to inhibit the enzyme COX-2. When inflammation occurs, COX-2 is able to enhance the prostanoid production [29]. As reported in Table 2, surprisingly, s-EPSs showed no significant differences compared with ibuprofen used as positive control when tested at the same concentration, thus suggesting a new role of s-EPSs in inflammation control.

### 3.2. Phycoerythrin Characterization

#### 3.2.1. Phycoerythrin Biocompatibility on Immortalized Eukaryotic Cells

Following biomass lysis, phycoerythrin (PE) had a purity grade of 1.5 [10]. This value is considered as reagent-grade, thus indicating that the protein can be used as it is for food applications [30]. In order to verify the safety of the protein on eukaryotic cells, an MTT assay was performed by comparing the crude extract with the purified protein (purity grade of four). The results of the MTT assay, reported in Figure 4, clearly show that only pure PE was fully biocompatible with both cell lines (Figure 4B), while the crude extract exerted a dose-dependent toxicity (Figure 4A). These results clearly indicate that PE needs to be purified to a higher purity grade before being used on cell-based models, or, at least, that it cannot be applied when present in the extract at concentrations higher than a certain threshold (100 µg/mL).

#### 3.2.2. *In Vitro* Antioxidant Activity

*In vitro* analysis of the antioxidant activity of purified PE was carried out with the abovementioned experimental procedures. As reported in Table 3, purified PE was not able to scavenge the DPPH radical or chelate copper ions. However, it demonstrated a high capacity to scavenge the ABTS radical ion and to reduce ferric iron or chelate iron with considerably low IC_50_ values (0.072 ± 0.004 and 0.084 ± 0.012 µM, 0.084 ± 0.004 µM, respectively). Noteworthy, the purified PE IC_50_ values were about 160, 1000, and 600 times lower than the IC_50_ values obtained with the positive control molecules (Trolox, 12 ± 1 µM in the ABTS; BHT, 90 ± 4 µM in the FRAP; and EDTA, 51 ± 3 µM in the ICA).

#### 3.2.3. Cell-Based Antioxidant Activity of PE

Starting from the encouraging results obtained *in vitro*, purified PE was tested on the UVA-stressed HaCaT experimental system used for s-EPSs. Cells were treated with 2.5 μg/mL (10 nM) of purified PE for 2 h, and then oxidative stress was induced by UVA irradiation (100 J/cm^2^). At the end of irradiation, the intracellular ROS levels were evaluated. As shown in Figure 5, UVA induced a significant increase in intracellular ROS levels (black bars, 200%) compared with untreated cells (*p* < 0.001). When cells were treated with purified PE (grey bars), no increase in intracellular ROS levels was observed. Interestingly, when cells were incubated with purified PE prior to UVA exposure, an inhibition of the intracellular ROS production was observed.

The effect of purified PE on GSH and lipid peroxidation was also assessed. As shown in Figure 6, PE was able to fully protect cells from oxidative stress, as no alteration in either the GSH levels (Figure 6A) or in the lipid peroxidation levels (Figure 6B) was found when the cells were pretreated with purified PE prior to stress, thus confirming the protective effect of the protein against oxidative stress.

#### 3.2.4. *In Vitro* PE Anti-Inflammatory Activity

Purified PE was also able to inhibit COX-2 (Table 4) by about 75%, although the level of inhibition attained with ibuprofen 24 nM could not be achieved at any of the analyzed concentrations.

### 3.3. Effect of s-EPSs and Purified PE on Wound Healing

Finally, a scratch assay was carried out on HaCaT cells to test the ability of s-EPSs and purified PE to induce cell migration related to wound repairing. The results are reported in Figure 7 and in Table 5. In the absence of any treatment, the cells spontaneously migrated to induce the re-epithelialization. Interestingly, when the cells were treated with either s-EPSs or purified PE, a significant enhancement in the wound closure was observed after 24 h. Indeed, s-EPSs reduced the scratched area by 2.5 ± 0.17-fold and purified PE by 2.4 ± 0.1317-fold compared with untreated cells (1.80 ± 0.02-fold reduction).

## 4. Discussion

As the use of synthetic molecules is known to be harmful in the long run, the search for new natural compounds endowed with beneficial properties is urgent [31]. In this context, antioxidants from microalgae could represent an excellent alternative, but the costs of microalgae upstream and downstream processes are still too high [32].

We recently set up a cascade approach to recover four classes of molecules from *P. cruentum* culture: s-EPSs, PE, carotenoids, and saturated fatty acids. Among them, here, we evaluated the biological activity of s-EPSs and PE. S-EPSs were chosen as it is generally thought that polysaccharides with a high sulfated content have biological activities [33], such as antioxidant action [34]. It is known that antioxidant molecules can bind metal ions, forming metal–ion complexes. The presence of sulfate groups could increase the metal-binding capacity of the carbohydrates by donating an electron pair or by losing a proton, thus stabilizing the complex [35,36].

In agreement with the findings of Wang et al., we found that s-EPSs had no radical scavenging activity against DPPH, whereas they showed antioxidant activity in the ABTS assay, with IC_50_ values ranging from 6.59 to 8.92 mg/mL [37]. Interestingly, despite the low antioxidant activity observed *in vitro*, s-EPSs were active on a cell-based system at a concentration almost 600 times lower than that measured *in vitro*. This result is in agreement with literature, as it is well known that *in vitro* assays should not be compared with cell-based ones. Indeed, antioxidants provide their function by different mechanisms of action, so that bioavailability, stability, retention, or reactivity of the compound under test in a complex system, such as that of eukaryotic cells, cannot be either mimicked or evaluated *in vitro* [38]. Our results indicated that s-EPSs were able not only to inhibit the intracellular ROS production but also to prevent GSH depletion and lipid peroxidation.

Different is the case of PE, which was found to be a very powerful antioxidant agent *in vitro* and on a cell-based system. The ABTS assay was in line with that observed by Sonani on a PE from a different source (IC_50_ of 72 ± 4 nM vs. 101 nM, respectively) [39], whereas the PE prepared by this author had lower DPPH-scavenging (IC_50_ of 930 nM) and iron-chelating abilities (IC_50_ of 484 nM) than the purified PE prepared in this study. We hypothesize that the higher antioxidant activity measured in our experimental system may rely on the source or strain used. A different source may also affect the biocompatibility results: indeed, we found that only pure PE was biocompatible with eukaryotic cells, strongly suggesting the importance of purification of the protein for all the potential applications. Pure PE protected cells from UVA irradiation at a concentration in the low nanomolar range (10 nM). Generally, antioxidants prevent the generation of free radicals, which can significantly affect some physiological processes, including wound healing. In particular, ROS generation can damage tissues and slow down the regeneration process. The presence of antioxidants should counteract chronic inflammation and at the same time contribute to promoting tissue regeneration [40]. Considering that both s-EPSs and PE were able to inhibit one of the key enzymes in the inflammation process (COX-2) and to induce a significantly faster scratch closure compared with untreated cells, we can conclude that the bioproducts obtained by *P. cruentum* represent an excellent ingredient for new biomaterials, such as medical patches.

## 5. Conclusions

In this study, s-EPSs and PE, obtained from *P. cruentum* culture by a cascade approach described in a previous work [10], showed a remarkable antioxidant activity in a cell-based system, higher than that obtained by *in vitro* assays, thus suggesting that the reliability of *in vitro* assays has to be overhauled. Moreover, both molecules showed anti-inflammatory characteristics comparable with ibuprofen and a significant ability to promote cell proliferation.

## Figures and Tables

**Figure 1 antioxidants-12-00337-f001:**
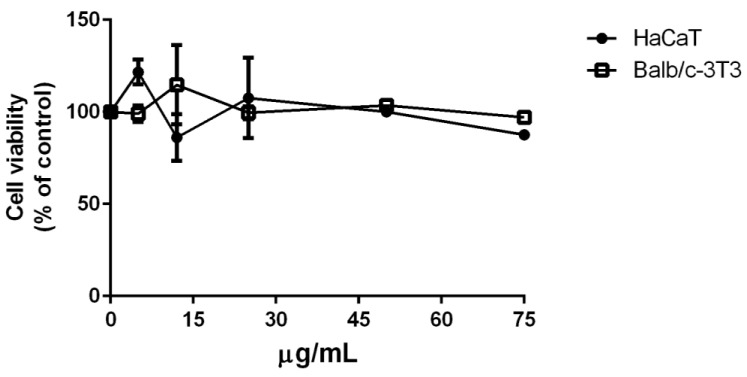
Effect of s-EPSs from *P. cruentum* on cell viability. Dose–response curves of HaCaT (black dots) and Balb/c-3T3 (empty squares) cells after 72 h of incubation with increasing concentrations of exopolysaccharides (5–75 µg/mL). Cell viability is reported as a function of s-EPS concentration.

**Figure 2 antioxidants-12-00337-f002:**
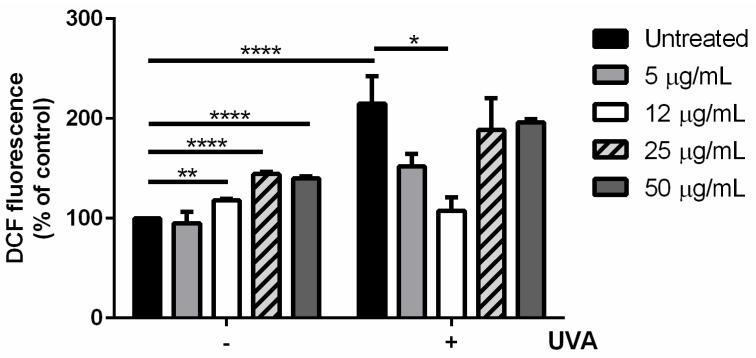
Antioxidant activity of s-EPSs on UVA-stressed HaCaT cells. Intracellular ROS levels were determined with DCFDA assay. Cells were preincubated in the presence of increasing amounts (from 5 to 50 µg/mL) of s-EPSs for 2 h prior to UVA irradiation (100 J/cm^2^). Results are expressed as percentages compared with untreated cells. Black bars refer to untreated cells; light grey bars refer to cells incubated with 5 µg/mL of s-EPSs; white bars refer to cells incubated with 12 µg/mL; dashed bars refer to cells incubated with 25 µg/mL; dark grey bars refer to cells incubated with 50 µg/mL of s-EPSs in the absence (−) or presence (+) of UVA stress. Data shown are means ± S.D. of three independent experiments. * indicates *p* < 0.05, ** indicates *p* < 0.01, and **** indicates *p* < 0.001.

**Figure 3 antioxidants-12-00337-f003:**
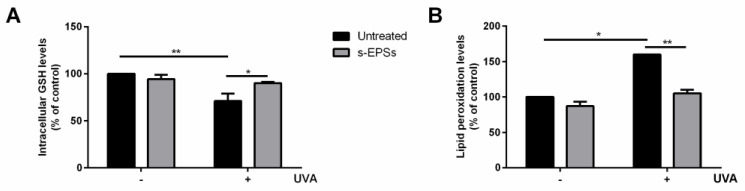
Protective effect of s-EPSs on HaCaT cells. Intracellular GSH levels were determined with a DTNB assay (**A**) and lipid peroxidation levels were determined with a TBARS assay (**B**). Cells were preincubated in the presence of 12 µg/mL of s-EPSs for 2 h prior to UVA irradiation (100 J/cm^2^). GSH and lipid peroxidation levels were measured 90 min after UVA irradiation. Black bars refer to untreated cells, and grey bars refer to cells incubated with s-EPSs, in the absence (−) or in the presence (+) of UVA stress. Values are expressed as percentages compared with untreated cells. Data shown are means ± S.D. of three independent experiments. * Indicates *p* < 0.05; ** indicates *p* < 0.01.

**Figure 4 antioxidants-12-00337-f004:**
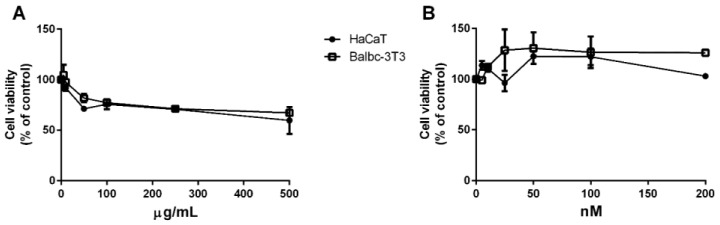
Biocompatibility of total extract (**A**) and purified PE (**B**) on eukaryotic cells. Dose–response curves of HaCaT (black dots) and Balb/c-3T3 cells (empty squares) after 72 h of incubation with increasing concentrations of total extract (**A**) and purified PE (**B**). Cell viability was assessed with an MTT assay and is reported as a function of extract/protein concentration.

**Figure 5 antioxidants-12-00337-f005:**
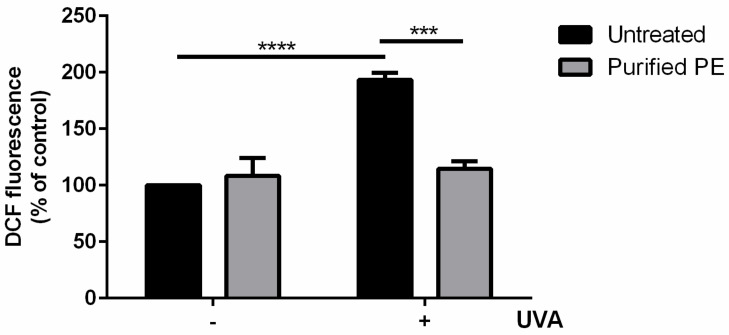
Protective effect of purified PE on UVA-stressed HaCaT cells. Intracellular ROS levels were determined with DCFDA assay. Cells were preincubated in the presence of 10 nM of purified PE (grey bars) for 2 h prior to UVA irradiation (100 J/cm^2^). Black bars refer to untreated cells in the absence (−) or in the presence (+) of UVA stress. Values are expressed as percentages compared with untreated cells. Data shown are means ± S.D. of three independent experiments. *** indicates *p* < 0.005; **** indicates *p* < 0.001 with respect to UVA-treated cells.

**Figure 6 antioxidants-12-00337-f006:**
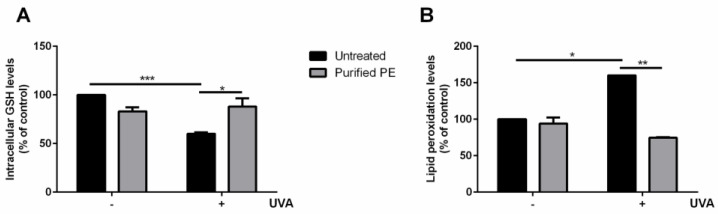
Analysis of intracellular GSH and lipid peroxidation levels on HaCaT cells. Cells were preincubated with 10 nM of purified PE for 2 h before UVA irradiation (100 J/cm^2^). (**A**) determination of intracellular GSH levels; (**B**) analysis of lipid peroxidation levels. In both experiments, measurements were recorded 90 min after UVA-induced stress. Values are expressed as a percentage compared with control (i.e., untreated) cells. Data shown are means ± S.D. of three independent experiments. * indicates *p* < 0.05, ** indicates *p* < 0.01, and *** indicates *p* < 0.005.

**Figure 7 antioxidants-12-00337-f007:**
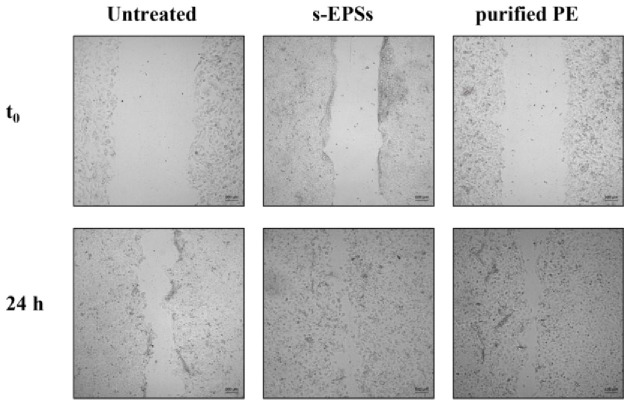
Effect of s-EPSs and purified PE on wound healing. Confluent HaCaT cells were scratched and treated with either 12 µg/mL s-EPSs or 10 nM purified PE for 24 h. Optical microscopy images were acquired at 10× magnification at the beginning (t_0_) and end (24 h) of the incubation.

**Table 1 antioxidants-12-00337-t001:** *In vitro* antioxidant and chelating activity of s-EPSs. Results are expressed as percentage of inhibition. The concentration evaluated is referred to the final concentration of s-EPSs or positive control used in the well.

Test	Concentration (µg/mL)	s-EPSs Activity (%)	C+ Activity (%)
FRAP	120	34 ± 3	97 ± 1
ABTS	25	2 ± 2	98 ± 2
DPPH	50	1 ± 1	52 ± 1
ICA	55	66 ± 3	91 ± 1
CCA	45	9 ± 3	91 ± 2

**Table 2 antioxidants-12-00337-t002:** *In vitro* s-EPSs anti-inflammatory activity.

	Concentration (µg/mL)	Inhibition (%)
s-EPSs	167	77 ± 8
Ibuprofen	167	99 ± 1

**Table 3 antioxidants-12-00337-t003:** *In vitro* antioxidant and chelating activity of purified PE. Results are expressed as IC_50_ values, µM.

Test	Purified PE	Positive Control
	IC_50_ (µM)	
ABTS	0.072 ± 0.004	12 ± 1
DPPH	>0.27	29 ± 2
FRAP	0.084 ± 0.012	90 ± 4
ICA	0.084 ± 0.004	51 ± 3
CCA	>0.1	63 ± 2

**Table 4 antioxidants-12-00337-t004:** COX-2 inhibition by purified PE.

Sample	Concentration (nM)	Inhibition (%)
Phycoerythrin	27	75 ± 8
	10	72 ± 8
Ibuprofen	24	96 ± 1

**Table 5 antioxidants-12-00337-t005:** Reduction of area (fold) of wound closure upon 24 h of incubation with either s-EPSs or purified PE. Data shown are means ± S.D. of three independent experiments. For each experiment, at least 10 images were acquired. * indicates *p* < 0.05.

Sample	Reduction in Area (Fold)
Untreated	1.80 ± 0.02
s-EPSs	2.50 ± 0.17 *
Purified PE	2.40 ± 0.13 *

* = *p <* 0.05.

## Data Availability

The data presented in this study are available in the article.

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
