# Peer review of "Shedding Light on the Hidden Benefit of Porphyridium cruentum Culture"

_antioxidants, 2023, doi:10.3390/antiox12020337_

Round 1

Reviewer 1 Report

The manuscript concerns an interesting issue related to characterization of algae based bioactive compounds. The introduction gives an adequate update on published material and the objectives are well formulated. The experimental setup relies on standard methods generally applied to this type of studies. The discussion is supported by data, being several works of other researchers considered.

The article is overall well written, concise and, in my opinion, can be recommended for publication.

Author Response

We thank the Referee for the very nice comment.

Reviewer 2 Report

Submitted manuscript titled “Shedding light on the hidden benefit of Porphyridium cruentum culture” is based on the assessment of the antioxidant and inti-inflammatory activity of microalgae-derived products.

The work is well planned and structured, and the methodology and the results are clearly described, even if the quality of some figures has to be improved.

The number of references is not so high, but most of them were published in the past 7-8 years, suggesting that the results of the present paper are relatively new and have the potential to be published. However, the authors could make an effort to further improve the introduction with additional information regarding the biological activity of this microalga (there are several papers available in literature). The discussion is short and concise (as it always should be!), and I agree with the hypotheses made by the authors.

I only recommend minor revisions, and my suggestions to improve this manuscript are listed below.

-          In my opinion, § 2.1 could be deleted and the only information provided should be inserted elsewhere in the M&M section

-          Line 55, “from the exhausted culture medium”: please clarify. Probably the authors mean that the cultures were analysed in the stationary phase. Please add information about culture volumes/concentrations, even if they are the same of ref. 10.

-          Please improve the quality of Fig. 4 and Fig. 7.

-          There are some typing errors (line 46: a capital letter, line 80: assay must be replaced by assays). Line 46: I prefer the full words to acronyms, even if their meaning is clear

Author Response

We thank the Referee for the nice comments and valuable suggestions. We changed the text according to his/her suggestions, where possible. Figures quality was improved.

Reviewer 3 Report

The manuscript antioxidants-2181189 "Shedding light on the hidden benefit of Porphyridium cruentum culture" provides an up-to-date and important information on the reliable sources of natural compounds with different activities.

The authors evaluated the antioxidant and anti-inflammatory activity of sulphated exopolysaccharides (s-EPSs) and Phycoerythrin (PE), two molecules naturally produced by the red marine microalga Porphyridium cruentum (CCALA415).

This data is relevant for a broad public since addresses a topical issue. It is perfectly clear to anyone familiar with this field of work that it had to be a great effort for authors to plan and conduct such a study, and I have a great appreciation for this. The manuscript is written in clear language and the background provides sufficient literature review. Overall, a good read.

After a very short revision of the text I fully support the publication of this paper in Antioxidants.

Author Response

We really thank the Referee for understanding and appreciating our efforts.

We fully revised the manuscript.